# The Potential of Gut Microbiota in Prediction of Stroke-Associated Pneumonia

**DOI:** 10.3390/brainsci13081217

**Published:** 2023-08-17

**Authors:** Zhongyuan Li, Mengmeng Gu, Huanhuan Sun, Xiangliang Chen, Junshan Zhou, Yingdong Zhang

**Affiliations:** 1Department of Neurology, Nanjing First Hospital, Nanjing Medical University, No. 68, Changle Road, Nanjing 210006, China; njmulzy@yeah.net (Z.L.); chenxl@njmu.edu.cn (X.C.);; 2Department of Neurology, Nanjing Yuhua Hospital, Nanjing 210039, China

**Keywords:** gut microbiota, stroke-associated pneumonia, stroke, short-chain fatty acids, predictor

## Abstract

Background: Stroke-associated pneumonia (SAP) is a common stroke complication, and the changes in the gut microbiota composition may play a role. Our study aimed to evaluate the predictive ability of gut microbiota for SAP. Methods: Acute ischemic stroke patients were prospectively enrolled and divided into two groups based on the presence or absence of SAP. The composition of gut microbiota was characterized by the 16S RNA Miseq sequencing. The gut microbiota that differed significantly between groups were incorporated into the conventional risk scores, the Acute Ischemic Stroke-Associated Pneumonia Score (AIS-APS), and the Age, Atrial fibrillation, Dysphagia, Sex, Stroke Severity Score (A2DS2). The predictive performances were assessed in terms of the area under the curve (AUC), the Net Reclassification Improvement (NRI), and the Integrated Discrimination Improvement (IDI) indices. Results: A total of 135 patients were enrolled, of whom 43 had SAP (31%). The short-chain fatty acids (SCFAs)-producing bacteria, such as *Bacteroides*, *Fusicatenibacter*, and *Butyricicoccus*, were decreased in the SAP group. The integrated models showed better predictive ability for SAP (AUC = 0.813, NRI = 0.333, *p* = 0.052, IDI = 0.038, *p* = 0.018, for AIS-APS; AUC = 0.816, NRI = 0.575, *p* < 0.001, IDI = 0.043, *p* = 0.007, for A2DS2) in comparison to the differential genera (AUC = 0.699) and each predictive score (AUC_AISAPS_ = 0.777; AUC_A2DS2_ = 0.777). Conclusions: The lower abundance of SCFAs-producing gut microbiota after acute ischemic stroke was associated with SAP and may play a role in SAP prediction.

## 1. Introduction

Acute ischemic stroke is a leading cause of disability and mortality, representing a major global health burden [1]. After a stroke, ischemic injury not only activates the neuroinflammation in the brain, but also results in an imbalance of the autonomic nervous system, which leads to systemic immunodepression [2,3]. Immunodepression mainly includes the transformation of T cells from pro-inflammatory T-helper (Th) 1 type to anti-inflammatory Th2 type, and the lymphocytosis in blood, spleen, and lymph nodes [4]. Immunodepression paves the way for post-stroke infections. The most common post-stroke infections are pneumonia and urinary tract infections, both reported with a percentage of 10% [5].

Stroke-associated pneumonia (SAP) is the most notable post-stroke infection for its significant association with poor stroke outcome [6,7,8]. SAP could increase stroke mortality and the length of in-hospital stay, affecting self-care ability [9,10]. Based on the known risk factors for SAP, such as age, dysphagia, and stroke severity [11,12], prediction models have been developed to identify patients at an increased risk for SAP, such as the Acute Ischemic Stroke-Associated Pneumonia Score [13] (AIS-APS), and the Age, Atrial fibrillation, Dysphagia, Sex, Stroke Severity Score [14] (A2DS2). Previous studies have proved the validity of AIS-APS and A2DS2 in predicting SAP [15,16]. In a recent meta-analysis, AIS-APS and A2DS2 showed moderate predictive accuracy for SAP [17]. To improve the discriminatory ability of the present models, more predictive risk factors of SAP need to be included.

Recent studies have demonstrated that a stroke could change the composition of patients’ gut microbiota, with an increase in opportunistic pathogens and a decrease in anti-inflammatory bacteria [18,19]. Furthermore, a stroke could reduce the transepithelial resistance of the gut barrier, disturb the immune homeostasis of the intestine, and promote the enrichment of proinflammatory bacteria [20]. The gut dysbiosis enhances the systemic inflammation in turn and induces the post-stroke infection [21]. Stanley et al. reported the activation of the sympathetic nervous system increased intestinal permeability after a stroke, leading to the translocation of the microbiota from the intestinal tract to the lung, ultimately resulting in post-stroke pneumonia [22]. The gut microbiota are thought to play a critical role in the occurrence and development of SAP, and several studies have proved it [23,24]. The feature of gut microbiota in stroke patients may be an essential predictor of SAP.

Therefore, this study aimed to (1) compare the composition of gut microbiota in acute ischemic stroke patients with and without SAP, (2) identify the patterns of altered gut microbiota in SAP patients, and (3) discern the improvement of predictive ability by integrating specific gut microbiota for SAP into conventional risk prediction models (i.e., AIS-APS and A2DS2).

## 2. Methods

### 2.1. Study Participants

This prospective observational cohort study was conducted in Nanjing First Hospital from May 2018 to June 2019. The inclusion criteria were as follows: (1) aged 50 years or older; (2) admitted within 72 h of symptom onset with a magnetic resonance imaging-confirmed diagnosis of acute anterior ischemic stroke; (3) have lived in Nanjing for at least six months. The exclusion criteria included: (1) use of antibiotics, probiotics, corticosteroids, or immunosuppressants in the last 30 days before admission; (2) a history of immune diseases, severe liver and kidney failure, or malignant tumors; (3) acute hemorrhagic stroke; (4) unavailable blood or stool samples. This study was approved by the Ethical Review Board of Nanjing First Hospital (Nanjing, China). The written informed consent was provided by all patients (or their immediate family members) before participating in this study.

SAP was diagnosed within seven days after stroke onset by two neurologists according to the 2015 Consensus Group criteria [25], with evidence of sputum culture or radiological signs of pulmonary infection on the chest-computed tomography. Participants who had SAP were included in the SAP group, while those without SAP were in the non-SAP (NSAP) group.

### 2.2. Data Collection

The basic information and medical histories were obtained from participants or their immediate family members through face-to-face conversations. All serum and stool samples were collected within 24 h of admission before the treatment with antibiotics or probiotics. The blood parameters were measured in the central laboratory of Nanjing First Hospital by the laboratory technicians who were blind to the clinical information. The hematological parameters including of white blood counts (WBCs), neutrophil, and lymphocyte counts were analyzed by Hematology Analyzer (BC-6900, Mindray, Shenzhen, China). The blood glucose and CRP were assayed by Fully Automated Biochemistry Analyzer (C16000, Abbott, Abbott Park, IL, USA).

The clinical subtype of stroke was determined by the Oxfordshire Community Stroke Project (OCSP) classification [26]. The etiology of stroke was described using the Trial of Org 10172 in Acute Stroke Treatment (TOAST) classification [27]. The neurological function was evaluated by modified Rankin Scale (mRS). The severity of stroke was assessed by the National Institute of Health Stroke Scale score (NIHSS). The level of consciousness was assessed by Glasgow Coma Scale (GCS), a 15-point scale composed of eye, verbal, and motor responses [28]. AIS-APS and A2DS2 were used to predict the risk of SAP. AIS-APS is a 34-point score based on demographics, medical history, pre-stroke mRS, stroke features, and admission glucose level [13]. A2DS2 score is a 10-point scale calculated as follows [14]: age ≥ 75 years = 1, atrial fibrillation = 1, dysphagia = 2, male sex = 1, NIHSS score of 0–4 = 0, NIHSS score of 5–15 = 3, and NIHSS ≥ 16 = 5. All clinical scores were evaluated by two experienced neurologists.

### 2.3. DNA Extraction and Sequencing

All 135 fecal samples underwent DNA extraction and sequencing in July 2019. Using the QIAamp^®^ DNA Stool Mini Kit (Qiagen, Hilden, Germany), the fecal DNA was extracted from stool samples according to the instructions. Briefly, the stool samples were lysed in argininosuccinate lyase buffer, and InhibitEX was used to adsorb impurities. Next, protease K was used to digest proteins, and DNA was purified with a two-step wash. After centrifugation, DNA was finally eluted from the spin column in the low-salt buffer. The concentration of DNA was calculated by measuring the absorbance of DNA eluate at 260 nm through the Nanodrop. The integrity was verified by 0.8% agarose gel electrophoresis.

The V3 to V4 hypervariable regions of the bacterial 16S rRNA gene were amplified by polymerase chain reaction (PCR) using the forward primer (5′-CCTACGGGNGGCWGCAG-3′) and the reverse primer (5′-GACTACHVGGGTATCTAATCC-3′) [29]. The amplified products were detected by gel electrophoresis and purified by the Agencourt AMPure XP Kit (Beckman Coulter, Brea, CA, USA). The index of purified products was performed in the 16S V3-V4 library. The Qubit@2.0 Fluorometer (Thermo Scientific, Waltham, MA, USA) and Agilent Bioanalyzer 2100 systems (USA) were used to evaluate the library quality. Using the 2 × 250 bp paired-end read protocol, high throughput sequencing was performed on the Illumina Miseq platform. This work was supported by the Shanghai Genesky Biotechnology Company (Shanghai, China).

### 2.4. Bioinformatics and Statistical Analyses

After quality filtering merging, UPARSE was used to cluster the raw reads into operational taxonomic units (OTUs) with a 97% similarity. All OTUs were classified by Mothur, according to Ribosomal Database Project (RDP) Release 9. Then, the Alpha diversities, including Chao, ACE, Shannon, Simpson, and Coverage index, were analyzed using Mothur. The Beta diversities were analyzed by permutational multivariate analysis of variance (PERMANOVA) and presented visually by principal coordinate analysis (PCoA). The Metastats and Linear discriminant analysis (LDA) Effect Size (LEfSe) was used to determine the microbial features between the two groups. The absolute values of logarithmic LDA score > 2 and *p*-value < 0.05 were considered statistically significant. The *p* values were adjusted with the Benjamini–Hochberg false discovery rate (FDR) correction for multiple testing. These analyses were performed on R version 3.4.3 (Vegan package).

The student *t*-test or Mann–Whitney *U* test was used to compare the statistical differences between the two groups for continuous variables. Fisher’s exact probability test or chi-squared test was applied for categorical variables. Also, Spearman’s rank correlation coefficient analysis was used to measure the correlation between gut microbiota and the prediction scores. The receiver operating characteristic curve (ROC) was performed to assess the predictive performance of gut microbiota. The Net Reclassification Improvement (NRI) and Integrated Discrimination Improvement (IDI) indices were generated to evaluate the improvement of predictive abilities after adding specific gut microbiota into AIS-APS and A2DS2. A *p*-value of <0.05 was considered to be statistically significant. All the statistical analyses were performed on SPSS 22.0 for Windows (IBM Inc., New York, NY, USA).

## 3. Results

### 3.1. Baseline Characteristics

We screened 732 patients with acute ischemic stroke and recruited 135 patients from May 2018 to June 2019. A total of 43 patients developed SAP (31.8%). Univariate analyses showed that patients in the SAP group were older than those in the NSAP group (71.5 vs. 66.3 years, *p* = 0.006). The SAP group, in comparison to the NSAP group, had a smaller percentage of male patients (51.2% vs. 75.0%, *p* = 0.006), and were more likely to have atrial fibrillation (44.2% vs. 15.2%, *p* < 0.001), dysphagia (39.5% vs. 8.7%, *p* < 0.001), and speech disorders (86.0% vs. 66.3%, *p* = 0.017). Patients with SAP had higher levels of WBC (9.2 vs. 7.4 × 10^9^/L, *p* = 0.001), neutrophil counts (5.6 vs. 4.4 × 10^9^/L, *p* < 0.001), C-reactive protein (9.4 vs. 2.5 ug/mL, *p* < 0.001), and neutrophil-to-lymphocyte ratios (NLRs) (4.8 vs. 2.7, *p* < 0.001) compared to the NSAP patients. Furthermore, higher NIHSS scores (11.0 vs. 3.0, *p* < 0.001) and lower GCS scores (8–15 vs. 15–15, *p* < 0.001) on admission, higher AIS-APS scores (11.0 vs. 5.0, *p* < 0.001), and higher A2DS2 scores (5.0 vs. 1.0, *p* < 0.001) were found in the SAP group compared with the NSAP group (Table 1).

### 3.2. Altered Gut Microbiota in the SAP Patients

The α-diversity indices of gut microbiota showed no significant group differences (Figure 1), except for the Simpson index, which indicated a lower diversity of the microbial community in the SAP group than that of the NSAP group (*p* = 0.022). The results of PCoA (Appendix A) revealed that Bray distances (Coefficient of determination (*R*^2^) = 0.014, *p* = 0.007), Jaccard distances (*R*^2^ = 0.010, *p* = 0.035), and weighted uniFrac distances (*R*^2^ = 0.050, *p* < 0.001) were significantly different between the two groups.

There were 112 unique OTUs in the SAP group, 979 unique OTUs in the NSAP group, and 1511 shared OTUs in both groups, as shown in the Venn diagram (Appendix A). The phyla of *Firmicutes*, *Bacteroidetes*, and *Proteobacteria* comprised most of the gut microbiota community. The Metastats analysis showed that the relative abundance of phylum *Bacteroidetes* in the NSAP group was higher than in the SAP group (*p* = 0.020) (Appendix A). At the family level, the abundances of *Bacteroidaceae* (*p* = 0.007), *Veillonellaceae* (*p* = 0.029), and *Sutterellaceae* (*p* = 0.014) were relatively lower in the SAP group (Appendix A). At the genus level, *Bacteroides* (*p* = 0.006), *Coprococcus* (*p* = 0.032), *Fusicatenibacter* (*p* = 0.039), *Butyricicoccus* (*p* < 0.001), *Butyricimonas* (*p* = 0.046), and *Clostridium_XlVb* (*p* = 0.018) were more enriched in the NSAP group (Figure 2). Also, the results of LEfSe analysis and LDA score showed that the relative abundances of class *Bacilli*, order *Lactobacillales*, family *Corynebacteriaceae*, genus *Corynebacterium,* and the species *Clostridium innocuum* in the SAP group were higher than those in the NSAP group (Figure 3).

### 3.3. Gut Microbiota Correlated with SAP Predictive Scores

Both scores of AIS-APS and A2DS2 were higher in the SAP group. The analyses on their correlation with the 50 most common genera showed that the genus *Bacteroides* was negatively correlated with AIS-APS (*r* = −0.197, *p* = 0.02) and A2DS2 (*r* = −0.320, *p* < 0.001). The same results were obtained for genus *Fusicatenibacter* (*r* = −0.25, *p* = 0.003 in AIP-APS; r = −0.22, *p* = 0.010 in A2DS2), *Butyricicoccus* (*r* = −0.24, *p* = 0.004 in AIP-APS; *r* = −0.219, *p* = 0.010 in A2DS2), and *Clostridium_XlVb* (*r* = −0.230, *p* = 0.007 in AIP-APS; *r* = −0.254, *p* = 0.003 in A2DS2). The genus *Enterococcus* had a significant positive correlation with both AIS-APS and A2DS2 scores (*r* = 0.297, *p* < 0.001 in AIS-APS; *r* = 0.360, *p* < 0.001 in A2DS2), though it did not show any significant difference between the SAP and NSAP groups (Figure 4).

### 3.4. Predictive Performance of Gut Microbiota for SAP

We tested the predictive validity of the six genera with significant group differences in Metastats analysis, including *Bacteroides, Coprococcus, Fusicatenibacter, Butyricicoccus*, *Butyricimonas*, and *Clostridium-IVb*. The area under the curve (AUC) of the differential genera was 0.699 (Standard Error (SE): 0.046, 95% Confidence Interval (CI): 0.609–0.790), while the AUCs of AIS-APS (SE: 0.047, 95% CI: 0.684–0.870) and A2DS2 (SE: 0.050, 95% CI: 0.679–0.874) were both 0.777. After adding the differential genera into AIS-APS, the AUC of the integrated model was improved to 0.813 (SE: 0.040, 95% CI: 0.735–0.891, as seen in Figure 5a). The results of IDI showed that the predictive ability of the model was significantly improved (IDI = 0.038, 95% CI: 0.006–0.070, *p* = 0.018), while NRI showed a borderline significant improvement (Continuous NRI = 0.333, 95% CI: −0.003–0.700, *p* = 0.052, Table 2). The ability of A2DS2 to predict SAP was also significantly improved after adding the differential genera, with the AUC improved to 0.816 (SE: 0.041, 95% CI: 0.735–0.897, as seen in Figure 5b), NRI 57.5% (95% CI: 0.245–0.906, *p* < 0.001), and IDI 4.3% (95% CI: 0.012–0.075, *p* = 0.007, as seen in Table 2).

## 4. Discussion

In this prospective observational cohort study, we explored the baseline differences in gut microbiota between SAP and NSAP patients. The two groups had significant differences in α-diversity and β-diversity, showing lower bacteria richness in SAP patients. Specifically, the genera *Bacteroides, Coprococcus*, *Fusicatenibacter*, *Butyricicoccus*, *Butyricimonas*, and *Clostridium_XlVb* were less abundant in the SAP group, while the abundances of family *Corynebacteriaceae*, genus *Corynebacterium*, and species *Clostridium_innocuum* were higher in the SAP group. Furthermore, the six decreased genera in the SAP group compared to the NSAP group could improve the predictive ability of AIS-APS and A2DS2.

Previous studies have revealed an alteration in gut microbiota composition after a stroke, which can differ depending on the severity of the stroke [18,19,30]. As a stress event, the stroke will activate the sympathetic nervous system, hypothalamic–pituitary–adrenal axis, and enteric nervous system, increasing the permeability of the intestinal mucosal barrier and the translocation of gut microbiota [20,31]. With the close association between the gut and the lung, gut dysbiosis could influence pulmonary health [32]. Chen et al. reported that commensal gut microbiota played a vital role in immune defense against *Escherichia Coli* pneumonia by inducing the expression of Toll-like receptor 4 (TLR) and activation of nuclear factor κB [33]. By stimulating TLR, gut microbiota can regulate the immune response of respiratory mucosa against influenza virus infection [34]. As one of the most significant and abundant commensal flora in the human intestine, the phyla *Bacteroidetes* can stimulate the TLR signaling pathways and regulate Treg cells [35,36]. Our study detected the early changes in gut microbiota in SAP patients, particularly the decrease in phyla *Bacteroidetes,* which suggested impaired immune surveillance preceding SAP.

The genera *Bacteroides*, *Coprococcus*, *Butyricicoccus*, *Butyricimonas*, and *Fusicatenibacter*, of which the abundances were lower in SAP patients, are all SCFAs-producing bacteria [37,38,39,40]. Among them, *Fusicatenibacter* and *Butyricicoccus* were negatively correlated with AIS-APS and A2DS2. A similar decrease in SCFAs-producing bacteria has been detected in patients with ischemic stroke [18] and post-stroke infection [24]. SCFAs are essential for their anti-inflammatory properties and immunomodulatory effects, such as suppressing the production of pro-inflammatory Interleukin-6 (IL-6) and inducing the production of anti-inflammatory IL-10 [41,42,43,44]. For instance, as one of the SCFAs, acetate was reported to activate the interferon-β response to enhance the antiviral effect on pulmonary epithelial cells [45]. While butyrate, another member of SCFAs, was positively associated with a decreased risk of lower respiratory tract infection due to its anti-inflammatory properties [46]. The transplantation of SCFAs-producing bacteria could improve neurological dysfunction and reduce inflammation in aged mice with middle cerebral artery occlusion [47]. Furthermore, increasing the levels of SCFAs through symbiotic therapy could reduce the incidence of enteritis and ventilator-associated pneumonia in sepsis patients [48]. Counterintuitively, several well-known SCFAs-producing bacteria like *Roseburia* were not differential between the two groups in our study, which was inconsistent with a recent study [23]. We thought that this may be attributable to the discrepancy between two cohorts, such as stroke severity and dietary structures, which could influence gut microbiota.

The alteration in the gut microbiota after a stroke, which involves the growth of pro-inflammatory bacteria, could affect immune homeostasis [49]. Our study found that several pathogenic bacteria were enriched in the SAP group, including the family *Corynebacteriaceae*, the genus *Corynebacterium*, and the species *Clostridium innocuum*. The genus *Corynebacterium*, which belongs to the family *Corynebacteriaceae*, is a potential pathogen due to its ability to produce the diphtheria toxin [50]. *Clostridium Innocuum* was reported to be cytotoxic by triggering cell death through apoptosis, leading to a number of infectious diseases, such as intra-abdominal infection, pylephlebitis, and empyema [51]. Highet et al. described that *Clostridium Innocuum* could increase the susceptibility to infection [52]. Recently, the infection of *Clostridium Innocuum* was reported to be associated with severe gastrointestinal complications and extraintestinal infections [53]. The accumulation of pro-inflammatory bacteria in stroke patients suggested that the imbalance in the immunologic barrier [54] may be a cause of SAP.

Compared to NSAP patients, SAP patients scored higher on two conventional predictive scores of AIS-APS and A2DS2. These scores were negatively correlated with the SCFAs-producing bacteria *Bacteroides*, *Fusicatenibacter*, and *Butyricicoccus*. Moreover, the six decreased genera in the SAP group, *Bacteroides*, *Coprococcus*, *Fusicatenibacter*, *Butyricicoccus*, *Butyricimonas*, and *Clostridium-IVb*, were mostly SCFA-producing bacteria (except *Clostridium-IVb,* which contains both beneficial and pathogenic species [55]). Haak et al. established that the decline in SCFAs-producing bacteria was an independent predictor of post-stroke infection [24]. Xia et al. showed that decreased SCFAs-producing genus *Roseburia* and increased pathogenic bacteria were correlated with the risk of SAP [23]. Our study confirmed their observations and further proved the predictive value of the above bacteria when integrated into the predictive scores. When the group-differential genera were incorporated, the predictive performances of AIS-APS and A2DS2 were both enhanced. This would imply that an altered gut microbiota, especially a lower abundance of SCFA-producing bacteria, could be a predictor of SAP. The existing predictive scores mainly focus on the clinical symptoms and may ignore the change in homeostasis prior to SAP onset [17]. Incorporating altered gut microbiota at baseline may pioneer new ways of early predictive models of SAP after further clinical verification.

Interestingly, we found that the relative abundance of the genus *Enterococcus* was significantly correlated with AIS-APS and A2DS2. *Enterococcus* was thought to be one of the opportunistic commensal bacteria associated with several infections like urinary tract infection, endocarditis, and biliary tract infection [56]. *Enterococcus* was also reported to be enriched in pneumonia patients after hypertensive intracerebral hemorrhage [57] and ischemic stroke [23], possibly due to the immunodepression after the stroke [58]. However, *Enterococcus* was not significantly different between SAP and NSAP groups in our cohort. This may be attributable to the regional differences between participants from previous studies, whose different diet structures may have affected the gut microbiota composition.

Our single-center study had several limitations. Firstly, we did not characterize the alterations in stool pH, inflammatory markers in stool, leaky gut markers, and metabolites such as SCFAs in the fecal samples. Secondly, diet has been demonstrated to be an essential factor influencing the composition of the gut microbiota [59]. Although we enrolled residents with presumably consistent dietary structures, the dietary intake data of each participant were not collected. Thirdly, we collected fecal samples at a single time point, and the dynamic changes in the gut microbiota were unavailable due to the limited funds. Fourthly, our study was an observational analysis without functional data to show causal links; hence, the results should be treated with caution until the precise mechanism of brain–gut–lung communication is revealed. Finally, due to the small sample size of this study, the reliability of the early prediction and diagnosis data needs to be further verified.

## 5. Conclusions

In conclusion, the baseline abundance of SCFAs-producing gut microbiota was decreased in patients who developed SAP within a week after the onset of acute ischemic stroke. The differential gut microbiota between SAP and NSAP added prognostic value to the conventional SAP risk scores of AIS-APS and A2DS2. This study emphasized the role of the altered post-stroke gut microbiota in predicting SAP, which merits attention in further clinical research.

## Figures and Tables

**Figure 1 brainsci-13-01217-f001:**
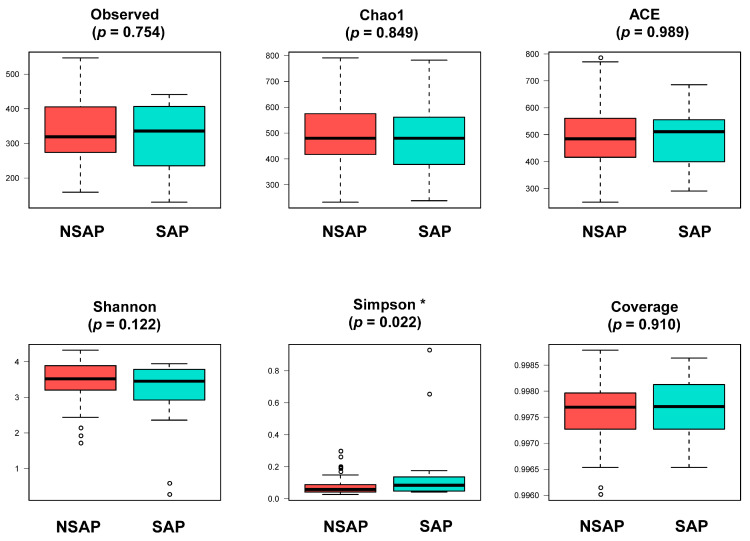
Comparison of α-diversity of gut microbiota between two groups. Abbreviation: SAP, stroke-associated pneumonia group; NSAP, non-SAP group. Note: * *p* < 0.05.

**Figure 2 brainsci-13-01217-f002:**
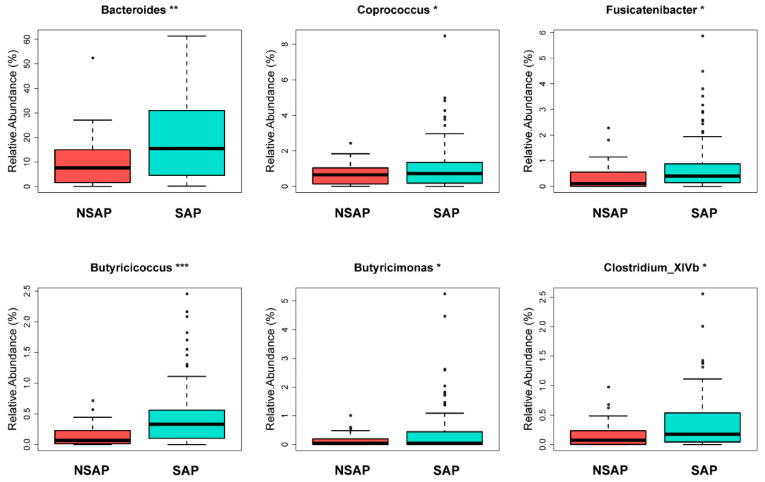
Gut microbiota with significantly different relative abundance between the two groups at genus level in Metastats analysis. Abbreviation: SAP, stroke-associated pneumonia; NSAP, non-SAP group. Note: * *p* < 0.05, ** *p* < 0.01, *** *p* < 0.001.

**Figure 3 brainsci-13-01217-f003:**
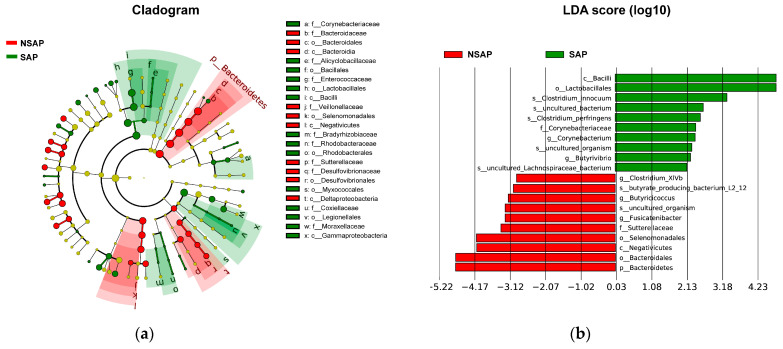
LEfSe of gut microbiota with significant difference between two groups. (**a**). Cladogram of differential bacteria; (**b**). Significantly differential bacteria with LDA scores > 2 between two groups. Abbreviation: LDA, Linear discriminant analysis; LEfSe, LDA Effect Size; SAP, stroke-associated pneumonia; NSAP, non-SAP group.

**Figure 4 brainsci-13-01217-f004:**
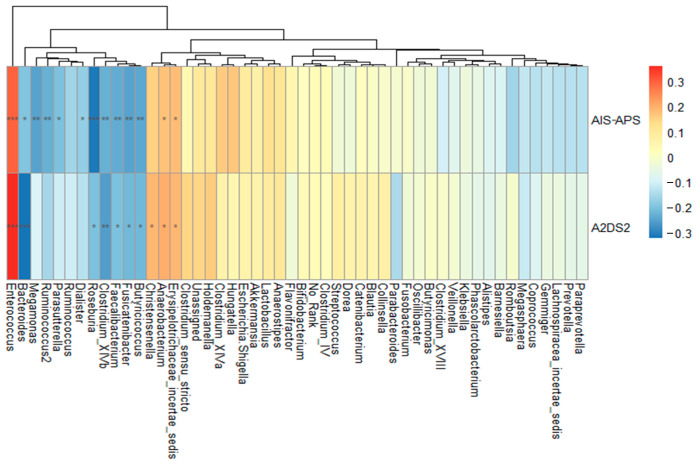
Spearman’s rank correlation analysis between the most 50 common genera and two predictive scores. Abbreviation: AIS-APS, the Acute Ischemic Stroke-Associated Pneumonia Score; A2DS2, the Age, Atrial fibrillation, Dysphagia, Sex, Stroke Severity Score. Note: * *p* < 0.05; ** *p* < 0.01; *** *p* < 0.001.

**Figure 5 brainsci-13-01217-f005:**
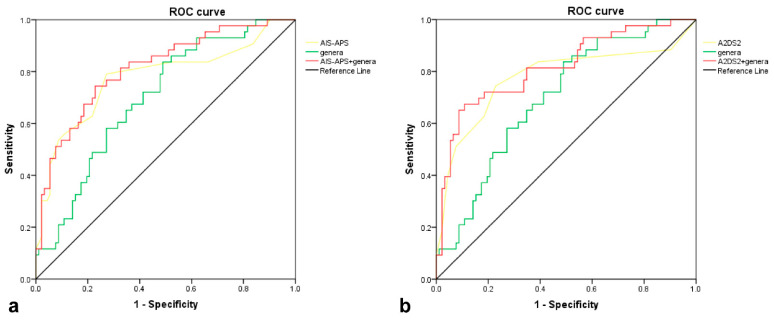
ROC curves of differential genera, 2 predictive scores, and the combination of genera and scores in predicting SAP. (**a**). ROC of genera (AUC = 0.699), AIS-APS (AUC = 0.777), and the combination of genera and AIS-APS (AUC = 0.813); (**b**). ROC of genera (AUC = 0.699), A2DS2 (AUC = 0.777), and the combination of genera and A2DS2 (AUC = 0.816). Genera consist of 6 bacteria at genus level, *Bacteroides*, *Coprococcus*, *Fusicatenibacter*, *Butyricicoccus*, *Butyricimonas,* and *Clostridium-IVb*, with significant difference between two groups in Metastats analysis. Abbreviation: ROC, receiver operating characteristic curve; AIS-APS, the Acute Ischemic Stroke-Associated Pneumonia Score; A2DS2, the Age, Atrial fibrillation, Dysphagia, Sex, Stroke Severity Score; SAP, stroke-associated pneumonia.

**Table 1 brainsci-13-01217-t001:** Baseline characteristics.

	Stroke-Associated Pneumonia		*p* Value
	Yes (N = 43)	No (N = 92)	
Age, mean (SD), y	71.5 (10.7)	66.3 (9.6)	0.006
Male, n (%)	22 (51.2)	69 (75.0)	0.006
Hypertension, n (%)	37 (86.0)	74 (80.4)	0.427
Diabetes mellitus, n (%)	15 (34.9)	35 (38.0)	0.723
Dyslipidemia, n (%)	28 (65.1)	53 (57.6)	0.407
Atrial fibrillation, n (%)	19 (44.2)	14 (15.2)	<0.001
Coronary heart disease, n (%)	5 (11.6)	8 (8.7)	0.591
History of stroke or TIA, n (%)	11 (25.6)	16 (17.4)	0.268
COPD, n (%)	1 (2.3)	2 (2.2)	0.687
Dysphagia, n (%)	17 (39.5)	8 (8.7)	<0.001
Speech disorders, n (%)	37 (86.0)	61 (66.3)	0.017
OCSP type, n (%)			<0.001
PACI or LACI	31 (72.1)	87 (94.6)	
TACI or POCI	12 (27.9)	5 (5.4)	
Fasting Glucose, median (IQR), mmol/L, (n = 130)	5.8 (5.0–7.9)	5.1 (4.4–6.6)	0.004
C-reactive protein, median (IQR), ug/mL, (n = 72)	9.4 (3.6–34.6)	2.5 (1.0–5.6)	<0.001
WBC, mean (SD), ×10^9^/L	9.2 (3.2)	7.4 (2.6)	0.001
Neutrophil, median (IQR), ×10^9^/L	5.6 (5.0–8.8)	4.4 (3.3–6.5)	<0.001
Lymphocyte, mean (SD), ×10^9^/L	1.5 (0.9)	1.7 (0.7)	0.147
NLR, median (IQR)	4.8 (3.1–8.4)	2.7 (2.0–4.2)	<0.001
Baseline NIHSS score, median (IQR)	11.0 (3.0–16.0)	3.0 (2.0–4.0)	<0.001
Baseline GCS, median (IQR)	15 (8–15)	15 (15–15)	<0.001
Baseline mRS, median (IQR)	0 (0–0)	0 (0–0)	0.216
Thrombolysis, n (%)	17 (39.5)	29 (31.5)	0.360
AIS-APS, median (IQR)	11.0 (7.0–20.0)	5.0 (3.0–7.0)	<0.001
A2DS2, median (IQR)	5.0 (2.0–7.0)	1.0 (1.0–2.0)	<0.001

Abbreviations: TIA transient ischemia attack; COPD chronic obstructive pulmonary disease; OCSP Oxfordshire Community Stroke Project classification; PACI partial anterior circulation infarct; LACI lacunar infraction; TACI total anterior circulation infarct; POCI posterior circulation infarct; SD standard deviation; IQR interquartile range; WBC, white blood cell count; NLR neutrophil-to-lymphocyte ratio; NIHSS National Institute of Health Stroke Scale score; GCS Glasgow Coma Scale; mRS modified Rankin Scale; AIS-APS the Acute Ischemic Stroke-Associated Pneumonia Score; A2DS2 the Age, Atrial fibrillation, Dysphagia, Sex, Stroke Severity score.

**Table 2 brainsci-13-01217-t002:** Reclassification statistics for the predictive ability of combination of differential genera and prediction scores for SAP.

		NRI (Continuous)		IDI	
Models	Variables	Estimate (95% CI)	*p* Value	Estimate (95% CI)	*p* Value
AIS-APS	+genera	0.333 (−0.003–0.700)	0.052	0.038 (0.006–0.070)	0.018
A2DS2	+genera	0.575 (0.245–0.906)	<0.001	0.043 (0.012–0.075)	0.007

The genera consist of 6 bacteria at genus level, *Bacteroides*, *Coprococcus*, *Fusicatenibacter*, *Butyricicoccus*, *Butyricimonas*, and *Clostridium-IVb*, with significant difference between two groups in Metastats analysis. SAP, stroke-associated pneumonia; NRI, Net Reclassification Improvement; IDI, Integrated Discrimination Improvement; CI, Confidence Interval; AIS-APS, the Acute Ischemic Stroke-Associated Pneumonia Score; A2DS2, the Age, Atrial fibrillation, Dysphagia, Sex, Stroke Severity Score.

## Data Availability

The raw sequence data of gut microbiota have been deposited in the Genome Sequence Archive [60] in National Genomics Data Center [61], China National Center for Bioinformation/Beijing Institute of Genomics and Chinese Academy of Sciences. The data have been publicly released at https://bigd.big.ac.cn/gsa/browse/CRA005250 (accessed on 14 August 2023), under accession number CRA005250.

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
