# Peer review of "The Potential of Gut Microbiota in Prediction of Stroke-Associated Pneumonia"

_brainsci, 2023, doi:10.3390/brainsci13081217_

Round 1

Reviewer 1 Report

The paper is well written and investigates an interesting topic.

I recommend publication after minor revision.

Did you also measure stool pH, inflammatory markers in stool, or leaky gut markers and SCFA concentrations? If yes, please add data. If not, please write, why you did not measure them and mention data, showing associations between these parameters and gut bacteria composition.

Please also discuss, whether alterations of glucose metabolism- which were evident between the two groups- might also reflect or be due to microbiota composition.

Please also comment, whether the number of comorbidities of patients was related to gut microbiota composition.

only minor typos, proven/proved

Author Response

Dear Reviewers:

Thank you for your comments concerning our manuscript entitled “The Potential of Gut Microbiota in Prediction of Stroke-Associated Pneumonia” (Manuscript ID: brainsci-2480226).

Those comments are all valuable and very helpful for revising and improving our paper, as well as the important guiding significance to our researches. We have studied comments carefully and have made correction which we hope meet with approval.

  1. Did you also measure stool pH, inflammatory markers in stool, or leaky gut markers and SCFA concentrations? If yes, please add data. If not, please write, why you did not measure them and mention data, showing associations between these parameters and gut bacteria composition.

Response: thanks for your suggestion.

We apologize that we did not measure the pH, inflammatory markers or SCFAs in stool samples. This part of the content was not considered in the design of this study for funding restrictions, and there is currently no way to supplement it. This is also one of the limitations of our study and we have described it in the limitation section of discussion part (Page10, Line 324-326, highlighted in yellow). In fact, we measured some inflammatory markers and microbial metabolites such as IL-10, IL-17 and TMAO in serum samples. However, these parameters were not different between SAP and NSAP group (shown in the following table), moreover, the association between the inflammatory/metabolic factors and the gut bacteria composition was beyond the scope of our work, so we didn’t mention it in our results part.

SAP

P value

Yes (N = 43)

No (N = 92)

IL-10, mean (SD), μg/l

51.5±16.5

48.3±18.1

0.322

IL-17, mean (SD), ng/l

81.3±17.9

79.0±17.1

0.475

TMAO, median (IQR), μmol/l

2.2(1.3-3.7)

2.8(1.5-4.2)

0.114

  1. Please also discuss, whether alterations of glucose metabolism which were evident between the two groups might also reflect or be due to microbiota composition.

Response: thanks for your suggestion.

In our study, the fasting glucose was higher in SAP group than NSAP group. It was indicated in previous studies that the alteration of glucose metabolism was due to the composition of gut microbiota, especially the decreased SCFAs-producing bacteria. SCFAs-producing bacteria, reported to be significantly decreased in type 2 diabetes patients (Nature. 2012;490(7418):55-60), were considered to be able to control blood glucose and improve insulin sensitivity (Nature 498: 99-103; Gastroenterology 143: 913-916 e917). Our previous study also suggested the negative correlation between SCFAs-producing bacteria Roseburia and fasting glucose in stroke patients (Front Cell Infect Microbiol. 2021;11:669322).

  1. Please also comment, whether the number of comorbidities of patients was related to gut microbiota composition.

Response: thanks for your question.

Our study collected a total of six comorbidities of stroke patients including hypertension, diabetes mellitus, dyslipidemia, atrial fibrillation, coronary heart disease, and COPD. The number of comorbidities was higher in SAP group than NSAP group (p = 0.047). Among them, only atrial fibrillation was significantly different between SAP and NSAP group. We explored the correlation between the number of comorbidities and the relevant abundance of the genera that were different between groups using Spearman correlation analysis, and found that the number of comorbidities was not correlated with the differential gut microbiota (shown below).

r

p

Bacteroides

-0.119

0.170

Coprococcus

-0.067

0.437

Fusicatenibacter

0.057

0.512

Butyricicoccus

-0.008

0.926

Butyricimonas

-0.023

0.788

Clostridium-IVb

-0.018

0.839

Reviewer 2 Report

Li et al. brain sciences, MDPI

Introduction:

·       is quite short. Check latest publications for more information.

Methods:

Several details are missing:

2.1.:

·       It reads like SAP is part of forth exclusion criterion. Please add paragraph.

2.1:

·       More information about approval would be good to have. Were participants asked?

2.3:

·       lane 85: please describe “according to the specifications” in more detail

·       Details about general evaluation is missing

·       Details about blood collection and serum analysis is completely missing

·       Glasgow coma score is not mentioned in methods

·       Check Table 1 of results and describe all methods used to receive this results in methods

Results: More details and explanations are needed. For scientists not working in the field, this is really hard to understand.

3.1:

·       “43 patients developed SAP” This reads like patients developed SAP after study recruitment, but they already had SAP when recruited, corrected?

3.2:

·       Figure S1: Please explain abbreviation PCoA

·       Lane 152-153: Compare SAP vs. NSAP and not vise versa

·       Figure 3: Text in figure is too small, please increase font size.

3.3.

·       First sentence in lane 172: This sentence seems redundant as the scoring systems were developed to detect SAP. Please rephrase.

·       Authors should also mention positive correlations shown in Figure 4 as these seem to be highly significant and thus relevant. Specifically, enterococcus is known to be able to induce several disorders. Results should also be discussed

3.4.

·       Figure 5: Coloring of groups is hard to distinguish, specifically in legend, please improve.

Discussion

·       Lane 223: these bacteria families are not mentioned in results, only in figures. Please add details to text of results if you want to discuss them.

References:

·       #13: Details are missing

·       The newest reference is from the year 2021 please update references by checking latest publications.

Reviewer’s summary:

Major changes are needed.

My #1 major concern is that authors did not mention the strongest significance of the whole manuscript (Fig. 4, Enterococcus), neither in Results nor in Discussion, why?

Discussion needs to be updated based on the latest literature.

English: Is ok but should be checked by native speaker.

Recommendation: Use ICMJE Recommendations to check your manuscript for completeness

Quality of English is good.

Author Response

Dear Reviewers:

Thank you for your comments concerning our manuscript entitled “The Potential of Gut Microbiota in Prediction of Stroke-Associated Pneumonia” (Manuscript ID: brainsci-2480226).

Those comments are all valuable and very helpful for revising and improving our paper, as well as the important guiding significance to our researches. We have studied comments carefully and have made correction which we hope meet with approval.

  1. Introduction: is quite short. Check latest publications for more information.

Response: thank you for your suggestion. We have expanded and renewed the introduction part with latest publications (Introduction part, highlighted in yellow).

  1. Methods: 2.1.: It reads like SAP is part of forth exclusion criterion. Please add paragraph.

Response: thanks for your suggestion.

We have added a new paragraph to describe SAP (Page 2, Line 81-83, highlighted in yellow).

  1. More information about approval would be good to have. Were participants asked?

Response: thanks for your suggestion.

Participants or their immediate family members were asked before participating in our study and all wrote informed consent, which is in line with the methods of our previous work (PMID: 35153980 and 34737970). This information has been included in the paper (Page 2, Line 77-80, highlighted in yellow).       

  1. Line 85: please describe “according to the specifications” in more detail.

Response: thanks for your suggestion.

In detail, the stool samples were lysed in ASL buffer firstly and the suspension was incubated at 70℃ for 5 minutes. The InhibitEX were used to absorb PCR inhibitors and DNA-degrading substances. After centrifugating, the impurities were precipitated and the DNA remained in the supernatant. The protein in the sample was digested with protease K with vortex and incubation at 70℃. The sample was then transferred to the QIAamp spin column where DNA was absorbed onto the QIAamp membrane after centrifugation. DNA was purified by a two-step wash and was finally eluted from the spin column in low-salt buffer. The concentration of DNA sample was calculated by measuring the absorbance of DNA eluate at 260nm. We have described the isolation of stool DNA briefly in our paper (Page 3, Line 106-111, highlighted in yellow).

  1. Details about general evaluation is missing

Response: thanks for your reminding.

The concentration of DNA was detected by measuring the absorbance of DNA eluate at 260nm through the Nanodrop and the integrity was checked by 0.8% aga-rose gel electrophoresis. We have added the description to the methods part (page 3, Line 109-111, highlighted in yellow).

  1. Details about blood collection and serum analysis is completely missing

Response: thanks for your reminding.

The blood samples were collected after within 24 hours of admission and measured at the hospital central laboratory with laboratory staff blinded to clinical data, the serum analysis of this study is within patients’ diagnosis diagram, which is part of clinical routine for stroke patients. We have added the description to the Methods part (Page 2, Line 89-90, highlighted in yellow).

  1. Glasgow coma score is not mentioned in methods

Response: thanks for your reminding.

Glasgow coma score (GCS) is a 15-point scale to assess the severity of coma including of eye opening, verbal response and motor response. We have added the definition of GCS in the methods part (Page 2-3, Line 95-97, highlighted in yellow).

  1. Check Table 1 of results and describe all methods used to receive this results in methods.

Response: Thanks for your suggestion.

We have checked our Table 1 and improved our methods part with all the descriptions of methods we missed, including of OCSP classification, TOAST classification and mRS (Page 2, Line 90-94, highlighted in yellow).

  1. 3.1 “43 patients developed SAP” This reads like patients developed SAP after study recruitment, but they already had SAP when recruited, corrected?

Response: thanks for your question.

We apologize for the misunderstanding we have caused. Indeed, the patients developed SAP after study recruitment. As our study is a prospective study and SAP was an outcome to be observed, we did not know if they would develop SAP when we recruited patients and collected their samples.

  1. 3.2 Figure S1: Please explain abbreviation PCoA

Response: Thanks for your suggestion.

PCoA, as the abbreviation of principal coordinate analysis, is the visual result of permutational multivariate analysis of variance (PERMANOVA) according to four distances that showed the differences of Beta diversities between two groups. We have added it to the legend of figureS1.

  1. 3.2 Line 152-153: Compare SAP vs. NSAP and not use “versus”

Response: thanks for your suggestion.

We have modified all the “versus” to “vs.” in the result part (Page 3-4, Line 148-156, highlighted in yellow)

  1. 3.2 Figure 3: Text in figure is too small, please increase font size.

Response: thanks for your suggestion.

We have increased the font size in Figure 3.

  1. 3.3 First sentence in line 172: This sentence seems redundant as the scoring systems were developed to detect SAP. Please rephrase.

Response: thanks for your suggestion.

We have changed the sentence “SAP predictive scores, AIS-APS and A2DS2, were higher in the SAP group” to “Both AIS-APS and A2DS2 were higher in the SAP group” (Page 6, Line 198, highlighted in yellow).

  1. 3.3 Authors should also mention positive correlations shown in Figure 4 as these seem to be highly significant and thus relevant. Specifically, enterococcus is known to be able to induce several disorders. Results should also be discussed

Response: thanks for your suggestion.

The positive correlations shown in Figure 4 included Enterococcus, Erysipipelotrichaceae_incertae_sedis and Anaerobacterium, and Enterococcus showed the strongest correlation. The genus Enterococcus was significantly positively correlated with both AIS-APS and A2DS2 (r = 0.297, p < 0.001 in AIS-APS; r = 0.360, p < 0.001 in A2DS2). Enterococcus was thought to be one of the opportunistic commensal bacteria which could be associated with several infections such as urinary tract infection, endocarditis and biliary tract infection. Enterococcus was also reported to be enriched in the pneumonia patients after hypertensive intracerebral hemorrhage and ischemic stroke, which could be associated with the immunodepression after stroke. However, Enterococcus was not significantly differential between SAP and NSAP groups in our study. We thought it was due to the regional difference between participants from different studies, whose different diet structure might affect the gut microbiota composition. We have described the positive correlations between Enterococcus and two prediction scores in the result part (page 7, Line 203-206, highlighted in yellow) and discussed accordingly (Page 10, Line 315-323, highlighted in yellow).

  1. 3.4 Figure 5: Coloring of groups is hard to distinguish, specifically in legend, please improve.

Response: thanks for your suggestion.

We have improved the coloring of groups and legends in Figure5.

  1. Discussion Line 223: these bacteria families are not mentioned in results, only in figures. Please add details to text of results if you want to discuss them.

Response: thanks for your suggestion.

We have added details to the results part that the relative abundances of class Bacilli, order Lactobacillales, family Corynebacteriaceae, genus Corynebacterium and species Clostridium innocuum in the SAP group were higher than those in NSAP group (Page 5, Line 184-187, highlighted in yellow).

  1. References: #13: Details are missing

Response: thanks for your suggestion.

We apologize that we missed the details of the reference. We have corrected it to “Ni, J., Shou, W., Wu, X. & Sun, J. Prediction of stroke-associated pneumonia by the A2DS2, AIS-APS, and ISAN scores: a systematic review and meta-analysis. Expert review of respiratory medicine 15, 1461-1472 (2021).” (Page 11, Line 387-388).

  1. The newest reference is from the year 2021 please update references by checking latest publications.

Response: Thanks for your suggestion.

We have updated our references with the latest publications in the year 2023.

  1. Major concern is that authors did not mention the strongest significance of the whole manuscript (Fig. 4, Enterococcus), neither in Results nor in Discussion, why?

Response: Thanks for your suggestion.

Though the genus Enterococcus was significantly positively correlated with both AIS-APS and A2DS2, it was not significantly different between SAP and NSAP groups. Our observation was different from previous studies showing an association between Enterococcus and pneumonia after stroke. This could be attributed to the regional difference between participants from different studies, whose different diet structure might affect their gut microbiota composition. Given the aim of this study was to identify the distinctive genera between SAP and NSAP, we didn’t mention it in our original manuscript; while in this revision, we have added it in our result (page 7, Line 203-206, highlighted in yellow) and discussion part (Page 10, Line 315-323, highlighted in yellow).   

  1. Discussion needs to be updated based on the latest literature.

Response: Thanks for your suggestion.

We have checked the discussion part and updated it with the newest publications (discussion part, highlighted in yellow).

  1. English: Is ok but should be checked by native speaker.

Response: Thanks for your suggestion.

We have carefully reviewed our article and corrected English grammar.

  1. Recommendation: Use ICMJE Recommendations to check your manuscript for completeness.

Response: Thanks for your suggestion.

The manuscript has been carefully checked through ICMJE Recommendations.

Reviewer 3 Report

The authors present a gut microbiota study of patients with stroke who develop PNA. Comparing to existing stroke PNA predictive scales they add some benefit for prediction. 

Overall very interesting study to open up this genera. 

Could the authors  comment on how they verify that patients have lived in Nanjing for at least six months, used antibiotics, probiotics, corticosteroids, or immunosuppressants in the last 30 days before admission; a history of immune diseases, severe liver and kidney failure, or malignant tumors. These are nearly impossible data to gather accurately on stroke patients. 

Author Response

Dear Reviewers:

Thank you for your comments concerning our manuscript entitled “The Potential of Gut Microbiota in Prediction of Stroke-Associated Pneumonia” (Manuscript ID: brainsci-2480226).

Those comments are all valuable and very helpful for revising and improving our paper, as well as the important guiding significance to our researches. We have studied comments carefully and have made correction which we hope meet with approval.

1.“Could the authors comment on how they verify that patients have lived in Nanjing for at least six months, used antibiotics, probiotics, corticosteroids, or immunosuppressants in the last 30 days before admission; a history of immune diseases, severe liver and kidney failure, or malignant tumors. These are nearly impossible data to gather accurately on stroke patients.”

Response: thanks for your suggestion.

All the demographic information and medical histories were collected from participants or their immediate family members through a face-to-face interview by two experienced neurologists. Most of our participants are local residents and we double-checked their medical records to check if they have certain medical histories or drugs taken in the last 30 days.

Round 2

Reviewer 2 Report

Authors significantly improved their manuscript but some changes are still needed:

Lane 89: which methods were used for serum analyses?

Lane 96: Please provide reference for Glasgow Coma Scale

Lane 325: "What is meant with stool PH? Is this pH?

English still lacks precision and quality needs to be checked. I recommend to ask a native speaker to edit it.

Language still lacks precision and quality of English needs to be checked. I recommend to ask a native speaker for editing.

Author Response

Dear Reviewers:

Thank you for your comments concerning our manuscript entitled “The Potential of Gut Microbiota in Prediction of Stroke-Associated Pneumonia” (Manuscript ID: brainsci-2480226).

We have studied comments carefully and have made correction which we hope meet with approval.

  1. Line 89: which methods were used for serum analyses?

Response: thanks for your suggestion.

The blood cell counts (WBC, neutrophils and lymphocytes), blood glucose and CRP were analyzed in our study. The blood cell counts were tested by Hematology Analyzer (BC-6900, Mindray, China). The blood glucose and CRP were assayed by Fully Automated Biochemistry Analyzer (C16000, Abbott, United States) . We have added the methods to our manuscript (Page 2, Line 88-92, highlighted in yellow).

  1. Line 96: Please provide reference for Glasgow Coma Scale

Response: Thanks for your reminder.

We have added the reference “Mehta, R. & Chinthapalli, K. Glasgow coma scale explained. BMJ (Clinical research ed.) 365, l1296 (2019)” for GCS (Page 12, Line 415, Reference 28).

  1. Line 325: "What is meant with stool PH? Is this pH?

Response: Thanks for your reminder.

It was our mistake and the “PH” have been modified to “pH” (Page10, Line 331, highlighted in yellow).

  1. English still lacks precision and quality needs to be checked. I recommend to ask a native speaker to edit it.

Response: Thanks for your reminder.

We have carefully checked our article and corrected our English. All the modifications were highlighted in yellow.